# IKKγ/NEMO Is Required to Confer Antimicrobial Innate Immune Responses in the Yellow Mealworm, *Tenebrio Molitor*

**DOI:** 10.3390/ijms21186734

**Published:** 2020-09-14

**Authors:** Hye Jin Ko, Yong Hun Jo, Bharat Bhusan Patnaik, Ki Beom Park, Chang Eun Kim, Maryam Keshavarz, Ho Am Jang, Yong Seok Lee, Yeon Soo Han

**Affiliations:** 1Department of Applied Biology, Institute of Environmentally-Friendly Agriculture (IEFA), College of Agriculture and Life Sciences, Chonnam National University, Gwangju 61186, Korea; hjngo0129@naver.com (H.J.K.); yhun1228@jnu.ac.kr (Y.H.J.); misson112@naver.com (K.B.P.); chang9278@naver.com (C.E.K.); mariakeshavarz1990@gmail.com (M.K.); hoamjang@gmail.com (H.A.J.); 2School of Biotech Sciences, Trident Academy of Creative Technology (TACT), Chandrasekharpur, Bhubaneswar, Odisha 751024, India; drbharatbhusan4@gmail.com; 3P.G. Department of Bio-Sciences and Bio-Technology, Fakir Mohan University, Nuapadhi, Balasore, Odisha 756089, India; 4School of Biotechnology and Life Sciences, College of Natural Sciences, Soonchunhyang University, 22 Soonchunhyangro, Shinchang-Myeon, Asan, Chungchungnam-do 31538, Korea; yslee@sch.ac.kr

**Keywords:** insect immunity, *Tenebrio molitor*, NF-κB transcription factor, antimicrobial peptides, RNAi

## Abstract

IKK*γ*/NEMO is the regulatory subunit of the IκB kinase (IKK) complex, which regulates the NF-κB signaling pathway. Within the IKK complex, IKK*γ*/NEMO is the non-catalytic subunit, whereas IKK*α* and IKK*β* are the structurally related catalytic subunits. In this study, *TmIKKγ* was screened from the *Tenebrio molitor* RNA-Seq database and functionally characterized using RNAi screening for its role in regulating *T. molitor* antimicrobial peptide (AMP) genes after microbial challenges. The *TmIKKγ* transcript is 1521 bp that putatively encodes a polypeptide of 506 amino acid residues. *Tm*IKKγ contains a NF-κB essential modulator (NEMO) and a leucine zipper domain of coiled coil region 2 (LZCC2). A phylogenetic analysis confirmed its homology to the red flour beetle, *Tribolium castaneum* IKKγ (*Tc*IKKγ). The expression of *TmIKKγ* mRNA showed that it might function in diverse tissues of the insect, with a higher expression in the hemocytes and the fat body of the late-instar larvae. *TmIKKγ* mRNA expression was induced by *Escherichia coli*, *Staphylococcus aureus*, and *Candida albicans* challenges in the whole larvae and in tissues such as the hemocytes, gut and fat body. The knockdown of *TmIKKγ* mRNA significantly reduced the survival of the larvae after microbial challenges. Furthermore, we investigated the tissue-specific induction patterns of fourteen *T. molitor* AMP genes in *TmIKKγ* mRNA-silenced individuals after microbial challenges. In general, the mRNA expression of *TmTenecin1*, *-2*, and *-4*; *TmDefensin1* and *-2*; *TmColeoptericin1* and *2*; and *TmAttacin1a*, *1b*, and *2* were found to be downregulated in the hemocytes, gut, and fat body tissues in the *TmIKKγ-*silenced individuals after microbial challenges. Under similar conditions, *TmRelish* (NF-κB transcription factor) mRNA was also found to be downregulated. Thus, *TmIKKγ* is an important factor in the antimicrobial innate immune response of *T. molitor*.

## 1. Introduction

Unlike vertebrates, insects rely on their innate defense mechanisms for pathogen surveillance and immunity. The production of antimicrobial peptides (AMPs) is the most important mechanism of innate immunity in insects. AMPs are induced through the activation of two key signaling pathways, namely the toll and immune deficiency (IMD) pathways. In humans and most other vertebrates, the mode of action of the toll and IMD pathways are clearly identified. In humans and mice, ten (TLR1–10) and twelve (TLR1–9 and 11–13) functional toll-like receptors (TLRs) are known, respectively. TLRs provide protection against a wide variety of pathogens by regulating signaling through adaptor proteins, including the myeloid differentiation factor 88 (MyD88), TIR-domain-containing adapter-inducing interferon-β (TRIF), TRIF-related adaptor molecule (TRAM), and toll-interleukin 1 receptor domain containing adaptor protein (TIRAP). The antiviral [1,2,3], antibacterial [4,5], antifungal [6], and antiparasitic [7,8] roles of TLRs have been characterized in vertebrates. However, the IMD pathway has been partially characterized in anticancer [9] and anti-inflammatory properties [10,11,12] and implicated in neuronal development [13]. Further, the toll and IMD pathways do not appear to share any intermediate components and regulate differential expression of AMP-encoding genes via distinct NF-κB-like transcription factors [14,15].

In the context of the humoral immune response, especially in reference to AMP induction against invading pathogens, the toll and IMD pathways have been partially characterized in insects. In *Drosophila*, the requirement of toll and IMD pathway-related genes against septic injury have been monitored using an oligonucleotide microarray [16]. Further, while peptidoglycan recognition protein (PGRP)-SC1 and -SC2 are required for recognizing Gram-positive bacteria to initiate signaling through the toll pathway, PGRP-LB is required for recognizing Gram-negative bacteria and initiates the IMD pathway [17]. Furthermore, PGRP-SD (similar to mammalian CD14) acts as an extracellular receptor for Gram-negative bacteria and recognizes *meso*-diaminopimelic acid-type peptidoglycan [18] In addition, two miRNAs, miR-9a and miR-981, negatively regulate the IMD pathway, thereby downregulating the expression of the antibacterial peptide diptericin [19]. In *Bombyx mori*, the expression patterns of 11 putative toll-related receptors and two toll analogs were analyzed using microarrays, which showed that *BmToll* and *BmToll-2* are related to embryogenesis; *BmToll-2* and -4 may have a function in gut immunity; and *BmToll-10*, -*11*, and *BmToLK-2* may be involved in sex-specific biological functions [20]. Further, while the toll pathway is triggered by the invasive fungal pathogen *Beauveria bassiana*, the IMD pathway is activated by the Gram-negative bacteria *Escherichia coli* and *Serratia marcescens* [21]. Most significantly, the extracellular serine protease cascade involved in the toll signaling pathway has been described in great detail, wherein the processing of active spätzle (toll ligand) from pro-spätzle is manifested by the spätzle-processing enzyme and negatively regulated by serpin-48 [22,23]. The downstream regulators of the toll signaling cascade, including the adaptor protein MyD88/Tube, the kinase Pelle, IκB-like protein Cactus, the NF-κB-like transcription factor dorsal-related immunity factor (Dif), and dorsal, have been known in *Drosophila* and *Tribolium* [24,25] but partially characterized in other insects. The information provides compelling evidence that the toll pathway is activated by soluble recognition molecules that trigger distinct proteolytic cascades converging on spätzle.

The inhibitor of nuclear factor kappa B kinase regulatory subunit gamma (IKKγ) encodes the regulatory subunit of the inhibitor of the kappa B kinase (IKK) complex that activates inflammation, immunity, cell survival, and other pathway genes. Together with IKKα and IKKβ, IKKγ is required for cytokine-mediated activation of the NF-κB pathway. IKKγ also transcriptionally represses the NF-κB pathway by competitively binding with the nuclear coactivator cAMP-responsive element binding protein [26]. Further, IKKγ responds to various stimuli by phosphorylation, sumoylation, and ubiquitination, which also positively or negatively regulate NF-κB pathway activation [27]. The knockdown of IKKγ in mouse embryonic fibroblasts leads to inactivation of the NF-κB pathway in response to tumor necrosis factor-α and lipopolysaccharide [28]. The *Drosophila* IKK complex (signaling activated by lipopolysaccharides and mediated by IMD) phosphorylates the downstream NF-κB factor, Relish, and transcriptionally initiates the expression of the AMP diptericin. Thus, the *Drosophila* IKKγ-dependent signaling complex is exclusively IMD-dependent and toll-independent [29].

The yellow mealworm beetle *Tenebrio molitor* has emerged as a convenient model for studying insect innate immunity in the last decade. Previously, we have described a three-step proteolytic cascade of the extracellular toll signaling pathway that is elicited against Gram-positive bacteria and fungi [22,30]. In the present study, we have discovered the downstream regulatory components of the toll signaling pathway and their function in AMP induction using RNA interference (RNAi) screens [31,32,33]. *TmToll-7* RNAi showed the downregulation of five AMP genes in *T. molitor* when challenged with *E. coli* [34]. Moreover, the *T. molitor IMD*, in the IMD signaling pathway, was found to confer resistance against the Gram-negative bacteria *E. coli* by regulating the expression of nine AMP genes [35]. Furthermore, the silencing of *TmRelish*, the downstream NF-κB gene of the IMD pathway, showed the pivotal functional regulation in the immune response against Gram-negative and Gram-positive bacteria [36,37]. In *T. molitor*, regulatory components of the pathway downstream of IMD are yet to be elucidated. In this study, we have functionally characterized *T. molitor IKKγ* (*TmIKKγ*) using an RNAi screen by elucidating its role in regulating *T. molitor* AMP genes after microbial challenges. Moreover, we have studied the mRNA expression of *TmRelish* (NF-κB factor in the IMD pathway), *TmDorsal isoform X1* (*TmDorX1*), and *TmDorX2* (NF-κB factor in the toll pathway) in *TmIKKγ* knockdown individuals to understand the crosstalk with NF-κB signaling in the *T. molitor* toll and IMD pathways.

## 2. Results

### 2.1. Gene Organization and cDNA Structure of TmIKKγ

The cDNA sequence of *TmIKKγ* was determined by a local tblastn search against the *T. molitor* genome and the expressed sequence tag database using the *T. castanuem* IKKγ (*Tc*IKK*γ)* protein sequence (EEZ99267.2) as a query. Homology mapping of *TmIKKγ* against the *T. molitor* DNA-Seq database (unpublished) revealed a single exon of 1521 bp, beginning with the translation start codon “ATG” and ending with the stop codon “TAA” (Appendix A). The coding region of *TmIKKγ* translated to a polypeptide of 506 amino acid residues (Figure 1).

### 2.2. Domain Architecture and Phylogenetics of TmIKKγ

Domain analysis using InterProScan and a conserved domain search in the NCBI database revealed two conserved domains for *Tm*IKK*γ*: the NF-κB essential modulator (NEMO) domain (between amino acids 106 and 172) and the leucine zipper domain of coiled coil region 2 (LZCC2) of NEMO (between amino acids 373 and 460) (Figure 1). A multiple sequence alignment of the *Tm*IKKγ amino acid sequence with the IKKγ of insect orthologs showed conservation at the level of the NEMO domain and LZ of domain CC2 of NEMO (Appendix A). To explore the evolutionary position of *Tm*IKKγ proteins, we constructed a phylogenetic tree using the IKKγ homologs from different insect species. *Tm*IKKγ was clustered with *Tc*IKKγ (Appendix A). The full-length amino acid sequences of *Tm*IKKγ and *Tc*IKKγ showed a high degree of conservation with amino acid sequences from the Asian longhorned beetle *Anoplophora glabripennis* NEMO isoform X2 (*Ag*IKKγ) and the Colorado potato beetle *Leptinotarsa decemlineata* NEMO isoform X2 (*Ld*IKKγ). The beetle cluster of IKKγ proteins was related with the lepidopteran IKKγ cluster, which included IKKγ amino acid sequences from *Bombyx mori*, *Spodoptera litura*, and *Plutella xylostella*. The IKKγ sequences from dipterans (including *Drosophila melanogaster* Kenny; *Dm*Kenny) and hymenopterans formed separate clusters. This represents a valid evolutionary lineage for *Tm*IKKγ, clustering within the coleopteran IKKγ amino acid sequences.

### 2.3. Developmental and Tissue-Specific Expression Patterns of TmIKKγ mRNA

We measured the mRNA expression levels of *TmIKKγ* at five different developmental stages (egg, larva (early larva [YL] and late larva [LL]), prepupa (PP), pupa (days 1–7 [P1–P7]), and adult (days 1–5 [A1–A5])) using qRT-PCR (Figure 2A). Our results showed that *TmIKKγ* mRNA expression levels were consistent in all the developmental stages, with higher expressions in the pupal and adult stages. A slight decline in expression was noticed in the late-larval, prepupal, and late-pupal (days 5–7) stages. Further, we examined the *TmIKKγ* mRNA levels in the larval and adult tissues of *T. molitor*. The results indicated that *TmIKKγ* mRNA was expressed in all the larval tissues examined, with higher expressions in the hemocytes and fat body (Figure 2B). In *T. molitor* adults (five days old), the expression levels of *TmIKKγ* mRNA were greater in the Malpighian tubules, integument, and fat body (Figure 2C).

### 2.4. Temporal Expression of TmIKKγ mRNA after Microbial Infection

We investigated the regulation of *TmIKKγ* mRNA expression after fungal (*C. albicans*) and bacterial (*E. coli* and *S. aureus*) infections by performing a time-course study, conducted over a period of 48 h. *TmIKKγ* expression was analyzed by qRT-PCR in the whole body, hemocytes, gut, and fat body tissues of *T. molitor* at intervals of 3, 6, 9, 12, and 24 h post-microbial infection (Figure 3). In the whole body, *E. coli* was found to be the strongest elicitor of *TmIKKγ* mRNA at 9 h post-infection (Figure 3A). Furthermore, *E. coli* was the strongest elicitor of *TmIKKγ* mRNA in the hemocytes (Figure 3B), gut (Figure 3C), and fat body tissue (Figure 3D) of *T. molitor* larvae, in comparison to *S. aureus* and *C. albicans*. Maximum induction of the transcript was recorded at 9 h (16-fold) in the fat body tissue, following *E. coli* infection.

### 2.5. TmIKKγ RNAi and Larval Mortality Assay

As *TmIKKγ* mRNA expression was induced after infection with *E. coli*, *S. aureus*, and *C. albicans*, we hypothesized that *TmIKKγ* could be a candidate for conferring resistance against bacterial and fungal infections. To test this hypothesis, we investigated the survival of *T. molitor* larvae after silencing *TmIKKγ* mRNA, followed by a microbial challenge (Figure 4). The RNAi efficiency of *TmIKKγ* mRNA was compared to that treated with the control double-stranded (ds)RNA (ds *enhanced green fluorescent protein (EGFP)*-treated group). We confirmed an efficient knockdown of *TmIKKγ* mRNA (about 99%) (Figure 4A). After confirming the knockdown of *TmIKKγ* mRNA, we challenged the ds*TmIKKγ* and ds*EGFP*-treated larvae with bacteria (*E. coli* and *S. aureus*) and fungi (*C. albicans*). The mortality of *T. molitor* larvae (ds*TmIKKγ* and ds*EGFP*-treated groups) in response to the microbial challenge was monitored for 10 days. We found that the ds*TmIKKγ*-treated larvae showed increased susceptibility to microbial infections. A clear trend was observed in *E. coli*-infected ds*TmIKKγ*-treated larvae, with 50% mortality compared with ds*EGFP*-treated larvae (Figure 4B). With the *S. aureus* infection, the mortality trend was unclear initially, but, at day 6 post-infection, there was a significant increase in mortality in ds*TmIKKγ*-treated larvae as compared with ds*EGFP*-treated larvae (Figure 4C). The mortality observed in *TmIKKγ-*silenced larvae with a *S. aureus* infection was considered significant, even though a high mortality rate in the *dsEGFP* control was perplexing and might be related to the overall health status of the larvae included in the group. Close to 50% mortality was observed in ds*TmIKKγ*-treated and *C. albicans*-infected larvae (Figure 4D). The results suggest the requirement of *TmIKKγ* mRNA against bacterial and fungal challenges in *T. molitor* larvae.

### 2.6. Effects of TmIKKγ RNAi on the Expression of AMPs

As *TmIKKγ* mRNA knockdown resulted in increased susceptibility of the larvae to bacterial and fungal infections, we wanted to understand if *TmIKKγ* is one of the important factors regulating AMP production. The transcriptional activation of 14 different AMP genes was studied in ds*TmIKKγ*-treated and microbial-challenged larvae compared to the control (ds*EGFP*-treated and microbial-challenged larvae). The results suggest that, of the 14 AMPs tested, the expression of ten AMP genes, including *TmTene1*, *2* and *4*; *TmDef1* and *2*; *TmCole1* and *2*; and *TmAtta1a*, *1b*, and *2*, were significantly downregulated by *TmIKKγ* RNAi in hemocytes after being challenged with *E. coli*, *S. aureus*, and *C. albicans* (Figure 5A,B,D–H,J–L). This suggests a positive regulatory effect of *TmIKKγ* on the ten AMP genes. Similarly, in the gut tissue, the expression of ten AMP genes, including *TmTene1*, *2*, and *4*; *TmDef1* and *2*; *TmCole1* and *2*; and *TmAtta1a*, *1b*, and *2*, significantly decreased in ds*TmIKKγ*-treated larvae when compared to ds*EGFP*-treated larvae after the microbial challenge (Figure 6A,B,D–H,J–L). The same repertoire of AMP genes was also positively regulated in the fat body tissue of ds*TmIKKγ*-treated larvae after the challenge with *E. coli*, *S. aureus*, and *C. albicans* (Figure 7A,B,D–H,J–L). The transcriptional activation of *TmTene3*, *TmCec2*, *TmTLP1*, and *TmTLP2* was not positively regulated by *TmIKKγ* RNAi.

### 2.7. Effects of TmIKKγ RNAi on NF-κB Genes

After discovering the regulatory effects of *TmIKKγ* RNAi on the activation of a specific set of *Tenebrio* AMP genes, we were interested in studying the expression of *Tenebrio* NF-κB genes in the toll (*TmDorX1* and *TmDorX2*) and IMD (*TmRelish*) signaling pathways in ds*TmIKKγ*-treated and microbially challenged larvae. The results showed that the mRNA levels of *TmRelish* and *TmDorX2* were dramatically decreased by *TmIKKγ* RNAi in the fat body tissue. The expression of *TmRelish* transcripts was significantly decreased by *TmIKKγ* RNAi in the hemocytes and gut. Further, *TmDorX2* mRNA was decreased by *TmIKKγ* RNAi in the gut (Figure 8A–C).

## 3. Discussion

The inhibitor of NF-κB (IκB) interacts with the Rel-homology domain of NF-κB and masks the nuclear localization signal of NF-κB. This prevents the translocation of NF-κB to the nucleus, retaining the inactive protein in the cytoplasm [38,39]. IκB kinases (IKKs) phosphorylate the IκB protein in response to various stimuli, leading to its ubiquitination and proteasomal degradation. This releases NF-κB factor to the nucleus, where it is needed for the transcriptional activation of AMPs [40]. IKKs are a part of the high molecular weight signalosome complex, where IKKα and IKKβ (also known as IKK1/CHUK and IKK2, respectively) are the two IκB kinases, and IKKγ (NEMO/NF-κB essential modulator/IKKAP-1/FIP-3) acts as the non-catalytic regulatory subunit [41]. In fact, IKKs serve as an adaptor between the receptor/associated protein complexes and the catalytic IKKα and IKKβ subunits [42].

In this study, we identified an *IKKγ* homolog in the yellow mealworm beetle *T. molitor* (*Tm*IKK*γ)* and characterized it at the molecular level. The mammalian homologs of IKKγ have been known in *D. melanogaster* (*Dm*Kenny) and *Crassostrea gigas* (*Cg*NEMO), among invertebrates [43,44]. The structural characteristics of *Tm*IKK*γ* include the NEMO domain and the LZCC2 of NEMO. Similar to mammals, an IKK binding region, NEMO ubiquitin binding domain, and zinc finger domain have been reported in *Cg*NEMO [38,45].

IKK*γ/*NEMO shows extensive phosphorylation regulating its function for the activation of the NF-κB signaling pathway. The phosphorylation of tyrosine 374 and serine 377 of IKK*γ/*NEMO suppresses NF-κB signaling in humans—mediated by tumor necrosis factor α and Kaposi’s sarcoma-associated herpesvirus (KSHV) FADD-like interleukin-1β converting enzyme (FLICE) inhibitory protein (KvFLIP), respectively [27].

Ubiquitination of IKK*γ/*NEMO at specific lysine residues is also found to be conserved, and site-directed mutagenesis studies have indicated their direct role in NF-κB activation [46]. *Dm*Kenny comprises the ubiquitin binding in the ABIN and NEMO domain (UBAN) and a LC3-interacting region (LIR) or Atg8-interacting protein (AIM) motif. The LIR motif is responsible for the interaction between Kenny and Atg8a and the selective degradation of Kenny by autophagy [47,48]. *Tm*IKK*γ* also displayed the relaxed LIR motif (xLIR) from amino acid positions 27 to 32 (ESFVVL pattern) and from amino acid positions 103 to 108 (SSYEEI pattern). The iLIR server (http://ilir.warwick.ac.uk/) also predicted the conventional LIR motif (WxxL) in *Tm*IKK*γ* from amino acid positions 5 to 10 (DPFVKV pattern), 174 to 179 (KLFNEL pattern), and 487 to 492 (NNYDSL pattern). This indicates that *Tm*IKK*γ* could interact with the autophagosomal membrane protein microtubule-associated protein 1 light chain 3 (Atg8/LC3).

The alignment of *Tm*IKKγ with IKKγ from orthologous species showed conservation at the domain level, indicating the presence of phosphorylation and ubiquitination sites, but the details need to be investigated in the future. Moreover, for the IKK complex, post-translational modifications at the level of catalytic subunits IKKα and IKKβ would also be crucial to understanding the positive or negative activation of the NF-κB signaling cascade in *T. molitor*. For the present study, we screened IKKα and IKKβ homologs from the *T. molitor* RNA-Seq database using *T. castaneum* sequences as the query. We were interested in understanding the involvement of *TmIKKγ* in the regulation of NF-κB genes, which are involved in the toll and IMD signaling pathways in *Tenebrio*, and the subsequent transcriptional control of AMP genes.

The induction of *TmIKKγ* mRNA expression in the whole body, hemocytes, gut, and fat body tissues of *T. molitor* after microbial challenges was an early indicator of the putative involvement of *TmIKKγ* in innate immunity. Further, RNAi experiments revealed that *TmIKKγ* is required for larval survival against the Gram-negative bacteria *E. coli*, the fungus *C. albicans*, and the Gram-positive bacteria *S. aureus*. This suggested that impairment in the larval survivability in the absence of *TmIKKγ* is possibly due to the susceptibility of the larvae to microbial infections. It is known that NF-κB activation downstream of IKKs is significant in mounting a humoral response against a plethora of biotic stimuli, including a diverse group of pathogens and parasites. In the *Drosophila* IMD pathway, TAK1 kinase complex phosphorylates the IKK complex that eventually phosphorylates Relish (NF-κB) to release the Rel homology domain to the nucleus and leads to the transcriptional activation of AMP genes [49,50].

In the present study, we tried to understand the transcriptional activation of fourteen *Tenebrio* AMP genes in the *IKKγ-*silenced *T. molitor* larval hemocytes, gut, and fat body tissues. In the hemocytes, gut, and fat body of *TmIKKγ*-silenced larvae the mRNA expression of ten, nine, and ten AMP genes were downregulated, respectively, when compared with ds*EGFP* injected larvae (negative control). AMP genes, such as *TmTene1*, *TmTene2*, *TmTene4*, *TmDef1*, *TmDef2*, *TmCole1*, *TmCole2*, *TmAtta1a*, *TmAtta1b*, and *TmAtta2*, were found to be downregulated in the hemocytes, gut, and fat body of *TmIKKγ*-silenced *T. molitor* larvae challenged with *E. coli*, *S. aureus*, and *C. albicans*. In a previous study, knockdown of the *T. molitor* IMD protein (central adaptor protein in the IMD pathway that receives signals through PGRP-LC and PGRP-LE), led to the downregulation of a similar set of AMPs (excepting *TmDef1*) after challenge with microbes. The suppression of AMP expression was more striking in the *TmIMD*-silenced larvae challenged with *E. coli*, suggesting the requirement of the transcript to mount a defense response against the pathogen [35]. Considering the IKK complex as a downstream protein complex in the IMD signaling pathway of *T. molitor* that transcribes AMP gene expression upon being microbially challenged, it was obvious to observe the regulation of the same set of AMPs in *TmIMD* and *TmIKKγ* knockdown models in separate studies.

Further, the increased susceptibility of *T. molitor* larvae to *E. coli* infection in a *TmToll* knockdown model was attributed to the downregulation of *TmTene1*, *TmDef1*, *TmDef2*, *TmCole1*, and *TmAtta2* mRNA expression [34]. This highlights the diversity of the *T. molitor* innate immune system in recognizing the polymeric *meso*-diaminopimelic acid-type peptidoglycan of *E. coli*, signaling through the NF-κB protein sequences (Dif and Relish orthologs) for the transcriptional activation of a specific set of AMPs [51]. We therefore were interested to understand the NF-κB component that regulates the expression of the ten AMP genes in *TmIKKγ* knockdown *T. molitor* tissues. Interestingly, the expression of *TmRelish* was significantly decreased by *TmIKKγ* RNAi, and the expression of other NF-κB genes, including *TmDorX1* and *-X2*, were slightly decreased, suggesting that these ten AMP genes were mainly regulated by the *TmIKKγ*-dependent IMD pathway. In contrast, mRNA levels for nine AMP genes and two NF-κB genes, including *TmRelish* and *TmDorX2*, were dramatically decreased by *TmIKKγ* RNAi in the gut, indicating that *Tm*IKKγ activates both the *Tm*Relish-dependent IMD pathway and *Tm*DorX2-dependent toll pathway in the gut. Interestingly, the expression of *TmDorX1* was drastically increased by *TmIKKγ* RNAi, suggesting that *Tm*IKKγ positively and negatively regulates innate immune responses. In *D. melanogaster*, a synergistic immune response elicited by the toll and IMD pathways has been reported [14]. Another report stated that both the toll and IMD pathways could be activated by Gram-positive bacteria and fungi in *D. melanogaster* [52]. Further, AMPs have been clustered in three independent groups in *T. castaneum*: AMPs regulated by the toll pathway alone, by the IMD pathway alone, and by the toll and IMD pathways [15,53]. In the present study, we observed that *Tm*IKKγ could activate both the *Tm*Relish-dependent IMD pathway and *Tm*DorX2-dependent toll pathway in *T. molitor*.

Given these observations, it is important to identify the α and β isoforms of IKK in *T. molitor* and to understand whether they work independently or in a complex to maintain the immune homeostasis. We also need to elucidate the putative role of *Tm*IKKγ/NEMO in selective autophagic degradation of the IκB kinase. In the future, it will be interesting to visualize the crosstalk between the toll and IMD signaling pathways with respect to the immune response of *T. molitor*. Further, a categorization of *Tenebrio* AMPs regulated by the toll and IMD pathways independently and by the both toll and IMD pathways in response to various groups of microorganisms could be crucial towards understanding the humoral response.

## 4. Materials and Methods

### 4.1. Insect Rearing

*T. molitor* larvae were reared at 26 ± 1 °C and 60% ± 5% relative humidity in an environmental chamber established in the laboratory under dark conditions. The larvae were fed with an artificial diet (1.1 g of sorbic acid, 1.1 mL of propionic acid, 20 g of bean powder, 10 g of brewer’s yeast powder, and 200 g of wheat bran in 4400 mL of distilled water autoclaved at 121 °C for 15 min). Healthy 10th to 12th instar larvae were used for all experiments.

### 4.2. Preparation of Microorganisms

The Gram-negative bacterium *E. coli* (strain K12), Gram-positive bacterium *Staphylococcus aureus* (strain RN4220), and the fungus *Candida albicans* were used for the immune challenge experiments. While *E. coli* and *S. aureus* were cultured in Luria-Bertani broth (MB cell, Seoul, Korea) at 37 °C, *C. albicans* was cultured in Sabouraud dextrose broth (MB cell, Seoul, Korea) at 37 °C. The microorganisms were harvested, washed twice in 1X phosphate-buffered saline (PBS pH 7.0), centrifuged at 1146 × *g* for 10 min, and resuspended in 1X PBS. Thereafter, concentrations were measured at 600 nm (OD_600_) using a spectrophotometer (Eppendorf, Hamburg, Germany). Finally, 1 × 10^6^ cells/μL of *E. coli* and *S. aureus* and 5 × 10^4^ cells/μL of *C. albicans* were used for the immune challenge studies.

### 4.3. Identification and in Silico Analysis of TmIKKγ

*TmIKKγ* sequence was retrieved from the *T. molitor* RNA-Seq (unpublished) and expressed sequence tag database. Local tblastn analysis was conducted using *Tribolium castanuem IKKγ* amino acid sequence (EEZ99267.2) as a query. For the gene structure of *TmIKKγ*, a local tblastn analysis was performed with *TmIKKγ* as the query and the *T. molitor* DNA-Seq database (unpublished) as the subject. Full-length cDNA and deduced amino acid sequences of *Tm*IKK*γ* were analyzed using the BLASTx and BLASTp algorithms at NCBI (https://blast.ncbi.nlm.nih.gov/Blast.cgi), respectively. The full-length open reading frame (ORF) sequence was formatted using UltraEdit64-bit text-editor (http://www.ultraedit.com). The gene-finding software FGENESH was used for the prediction of the *TmIKKγ* open reading frame (http://www.softberry.com/berry.phtml?topic=fgenesh&group=programs&subgroup=gfind). The domain architecture of the protein sequences was retrieved using the InterProScan domain analysis program (https://www.ebi.ac.uk/interpro/search/sequence-search). Multiple sequence alignment, percentage identity, and percentage distance of the *Tm*IKKγ amino acid sequence with representative IKKγ from other insects were analyzed by the ClustalX 2.1 program (www.clustal.org) [54]. Phylogenetic tree was constructed based on the amino acid sequence alignments using the neighbor-joining method (bootstrap trial set to 1000). The phylogram was analyzed using Tree Explorer view under the Molecular Evolutionary Genetics Analysis (MEGA) version 7.0 (https://www.megasoftware.net/) program [55].

### 4.4. Developmental, Tissue-Specific, and Immune Induction of TmIKKγ mRNA

For the characterization of *TmIKKγ* mRNA in various developmental stages (egg, young-instar larva, late-instar larva, prepupa, 1–7-day-old pupa, and 1–2-day-old adult); larval tissues (integument, gut, fat body, Malpighian tubules, and hemocytes); and adult tissues (integument, gut, fat body, Malpighian tubules, hemocytes, ovary, and testes) of the insect were collected. Total RNAs were isolated by Clear-S Total RNA extraction kit (Invirustech Co., Gwangju, Korea). Briefly, samples were collected in 1-mL guanidine thiocyanate RNA lysis buffer (20-mM EDTA, 20-mM MES buffer, 3-M guanidine thiocyanate, 200-mM sodium chloride, 40-μM phenol red, 0.05% Tween-80, 0.5% acetic acid glacial (pH 5.5), and 1% isoamyl alcohol in 50-mL D.W.) and homogenized using a bead-based homogenizer (Bertin Technologies, Bretonneux, France) for 20 s. Subsequently, the extract was centrifuged at 21,000× *g* for 5 min at 4 °C. The supernatant fraction was mixed with 1 volume of 99% ethanol and incubated at room temperature (RT) for 1 min. The samples were transferred into a silica spin column (Bioneer, Daejeon, Korea, KA-0133-1) and centrifuged at 21,000× *g* for 30 s at 4 °C to remove debris. After eliminating the genomic DNA contamination (treatment with DNase at 1 U/µL and incubation at 37 °C for 15 min at RT), the samples were washed with 3-M sodium acetate, followed by 80% ethanol. After centrifugation at 21,000× *g* for 2 min at 4 °C, DNase and RNase-free water was used to elute the total RNA. cDNA was synthesized using total RNA (2 µg) as the template, Oligo (dT)_12–18_ primers, and AccuPower^®^ RT PreMix (Bioneer, Daejeon, Korea), according to the manufacturer’s instructions.

For the immune induction of *TmIKKγ* mRNA, healthy *T. molitor* larvae were intra-abdominally injected with 1-μL suspension containing either 1 × 10^6^ cells of *E. coli* or *S. aureus* or 5 × 10^4^ cells of *C. albicans.* A similar volume of PBS was administered into a separate group of larvae, which served as the wounded control. Samples (whole body, hemocytes, gut, and fat body tissue) were collected at 3, 6, 9, 12, and 24 h post-injection. Total RNA isolation and cDNA synthesis were done as described before.

The expression of *TmIKKγ* mRNA was evaluated by performing relative quantitative PCR using AccuPower^®^ 2X GreenStar^TM^ qPCR Master Mix (Bioneer, Daejeon, Korea), synthesized cDNAs, and specific primers for *TmIKKγ* (Table 1). The cycling conditions included an initial denaturation at 95 °C for 5 min, followed by 40 cycles of denaturation at 95 °C for 15 s, and annealing and extension at 60 °C for 30 s. Relative quantitative PCR assays were performed on an AriaMx Real-Time PCR System (Agilent Technologies, Santa Clara, CA, USA), and the results were analyzed using AriaMx Real-Time PCR software (www.agilent.com). Each analysis was measured independently at least three times. The 2^−ΔΔCt^ method [56] was employed to analyze the *TmIKKγ* mRNA expression levels. The mRNA expression levels were normalized to *T. molitor* ribosomal protein L27a (*TmL27a*), which was used as an internal control. The results represent the mean ± SE of three biological replicates.

### 4.5. TmIKKγ RNAi

For RNAi-based functional characterization, dsRNA fragments of *TmIKKγ* were synthesized—*TmIKKγ* DNA fragments were amplified by PCR using gene-specific primers tailed with a T7 promoter sequence (Table 1). The primers were designed using the SnapDragon software (http://www.flyrnai.org/cgi-bin/RNAi_find_primers.pl) to prevent any cross-silencing effects. The first PCR reaction for *TmIKKγ* was carried out using AccuPower Pfu-PCR PreMix (Bioneer, Daejeon, Korea) with cDNA and specific primers for *TmIKKγ* (Table 1). The second PCR reaction was conducted with 100-fold dilution of the first PCR product; the final PCR was performed with primers tailed with the T7 promoter sequence and 100-fold dilution of the second PCR product. The PCR reaction was conducted under the following conditions: initial denaturation at 94 °C for 5 min, followed by 35 cycles of denaturation at 94 °C for 30 s, annealing at 53 °C for 1 min, and extension at 72 °C for 30 s on a MyGenie96 Thermal Block (Bioneer, Daejeon, Korea). The PCR products were purified using the AccuPrep^®^ PCR Purification Kit (Bioneer, Daejeon, Korea), which were then used to synthesize double-stranded RNA (dsRNA) using the EZ^TM^ T7 High Yield in vitro transcription kit (Enzynomics, Daejeon, Korea). Thereafter, 1 μg of PCR purified product was used as template DNA and mixed with 4 μL of 5× transcription buffer, 2 μL of 10× MgCl_2_, 2 μL of 100-mM DTT, 1 μL of RNase Inhibitor (40 U/μL), 1 μL of 100-mM rATP, 1 μL of 100-mM rGTP, 1 μL of 100-mM rCTP, 1 μL of 100-mM rUTP, and 1 μL of RNA polymerase. Subsequently, the reaction mixture was incubated at 42 °C for 3 h. The dsRNA product was purified using the PCI (phenol:chloroform:isoamyl alcohol) method, precipitated with 5-M ammonium acetate, and washed with 70% and 90% ethanol. The synthesized dsRNA was stored at −20 °C until further use.

For the knockdown of *TmIKKγ* mRNA, 1 μg of synthesized dsRNA of enhanced green fluorescent protein (EGFP) and *TmIKKγ* were injected into the *T. molitor* 10th–12th instar larvae. *EGFP* dsRNA synthesized from pEGFP-C1 plasmid DNA was used as a negative control for RNAi.

### 4.6. Mortality Assay

To measure mortality, 1 × 10^6^ cells/μL of *E. coli* and *S. aureus* and 5 × 10^4^ cells/μL of *C. albicans* were injected into *T. molitor* larvae, which *TmIKKγ* had silenced. Dead larvae were counted daily up to 10 d post-infection; ten insect larvae were used for each set of the mortality assay, and the experiments were triplicated.

### 4.7. Effects of TmIKKγ RNAi on the Expression of AMP and NF-κB Genes

To further characterize the function of *TmIKKγ* in the humoral innate immune response, the effect of *TmIKKγ* silencing by RNAi on the expression levels of fourteen AMP genes in the presence of the microbial challenge were investigated. After treating the 10th to 12th instar *T. molitor* larvae with *TmIKKγ* dsRNA, *E. coli* (1 × 10^6^ cells per larva), *S. aureus* (1 × 10^6^ cells per larva), or *C. albicans* (5 × 10^4^ cells per larva) were injected into the larvae, and the samples (from the hemocytes, gut, and fat body) were collected 24 h post-injection. As an injection control, 1× PBS was used. Expression levels of fourteen AMP genes, including *TmTenecin 1*, *2*, *3*, and *4* (*TmTene1*, *2*, *3*, and *4*); *TmDefensin 1* and *2* (*TmDef1* and *2*); *TmColeoptericin 1* and *2* (*TmCole1* and *2*); *TmAttacin 1a*, *1b*, and *2* (*TmAtta1a*, *1b*, and *2*); *TmCecropin 2* (*TmCec2*); and *TmThaumatin-like protein 1* and *2* (*TmTLP1* and *2*) were measured by qRT-PCR with AMP gene-specific primers (Table 1). In addition, the expression of *Tenebrio* NF-κB genes in the toll (*TmDorX1* and *TmDorX2*) and IMD (*TmRelish*) signaling pathways were examined by qRT-PCR with the NF-κB-specific primers listed in Table 1. Relative quantitative PCR was performed, as mentioned above, with AMP-specific primers.

### 4.8. Statistical Analysis

All experiments were carried out in triplicate, and the data were subjected to one-way analysis of variance. Tukey’s multiple range test was used to evaluate the differences between groups (*p* < 0.05).

## 5. Conclusions

The study identified, for the first time, the IKKγ isoform in the beetle *T. molitor*. *TmIKKγ* transcript expression was induced in the whole larva and other tissues of *T. molitor* after *E. coli*, *S. aureus*, and *C. albicans* challenges. The RNAi experiments suggested that *TmIKKγ* might be required for larval survival against microbial challenges by regulating AMPs through the innate immune signaling cascade mechanisms. The NF-κB factors in the IMD pathway (Relish) and the toll pathway (Dorsal isoform 2) were involved, as their transcripts were downregulated in *TmIKKγ*-silenced larval tissues after the microbial challenge (Figure 9). This study advances our understanding of the beetles’ immunity in response to pathogenic challenges.

## Figures and Tables

**Figure 1 ijms-21-06734-f001:**
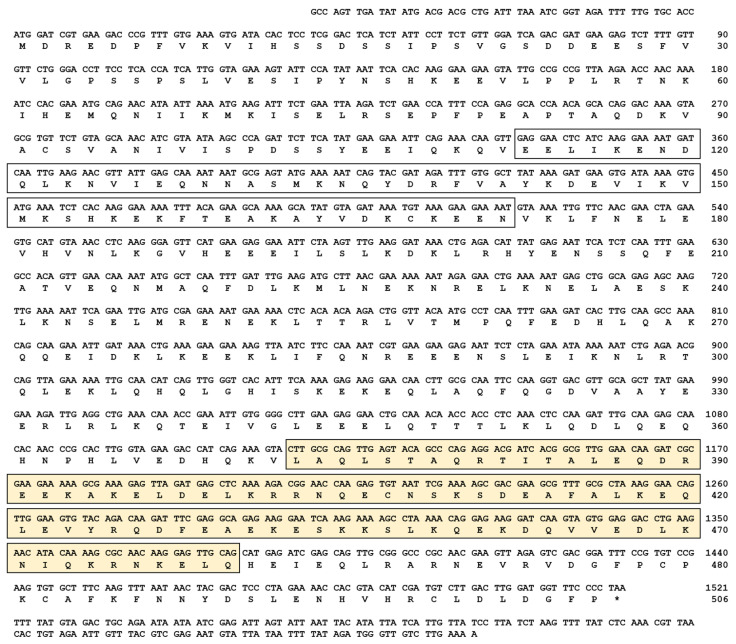
The complete nucleotide and deduced amino acid sequence of *Tenebrio molitor* (*Tm*)IKKγ. The nucleotides are numbered from the first base of the initiation codon to the stop codon. This represents the open reading frame region. The asterisk indicates the stop codon. The characteristic NF-κB essential modulator (NEMO) domain is enclosed by an open box, and the leucine zipper domain of coiled coil region 2 (LZCC2) of NEMO is enclosed by a yellow box.

**Figure 2 ijms-21-06734-f002:**
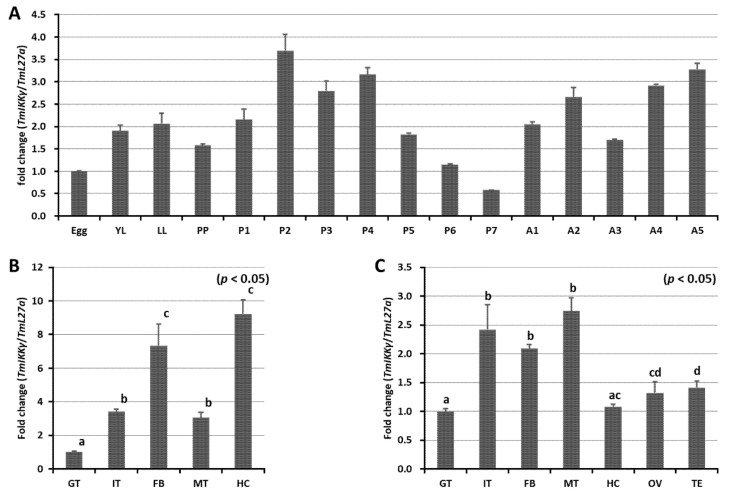
Developmental and tissue-specific expression of *TmIKKγ* mRNA using qRT-PCR. (**A**) Relative expression of *TmIKKγ* mRNA in the egg (Egg), early larva (YL), late larva (LL), prepupa (PP), pupa days 1–7 (P1–P7), and adult days 1–5 (A1–A5). (**B**) *TmIKKγ* mRNA expression in last-instar larvae. (**C**) Distribution of *TmIKKγ* mRNA in adult tissues. IT, integument; GT, gut; FB, fat body; HC, hemocytes; MT, Malpighian tubules; OV, ovary, and TE, testis. *L27a* from *T. molitor* was included as an internal control to normalize RNA levels between samples. Vertical bars represent standard errors (*n* = 3). Different letters above bars represent significant differences between groups.

**Figure 3 ijms-21-06734-f003:**
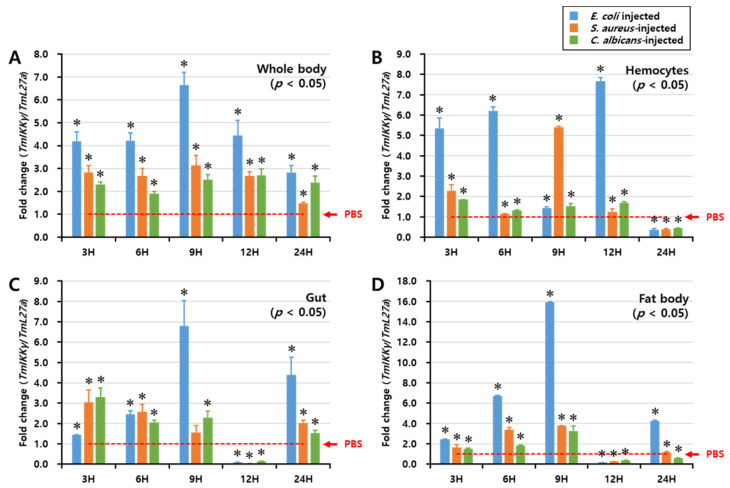
Temporal expression patterns of *TmIKKγ* mRNA in the whole body (**A**), hemocytes (**B**), gut (**C**), and fat body (**D**) of *T. molitor* post-challenge with *Escherichia coli*, *Staphylococcus aureus*, or *Candida albicans.* The expression was analyzed by qRT-PCR using *L27a* (*T. molitor*) as the internal control. For each time point, the expression level in the phosphate-buffered saline (PBS)-injected control (mock control) was set to 1; this is represented by a dotted line. Values represent the mean of triplicate experiments, mean ± SE (*n* = 20). Asterisk denotes significant differences from the mock control at 95% confidence level (Student’s *t*-test).

**Figure 4 ijms-21-06734-f004:**
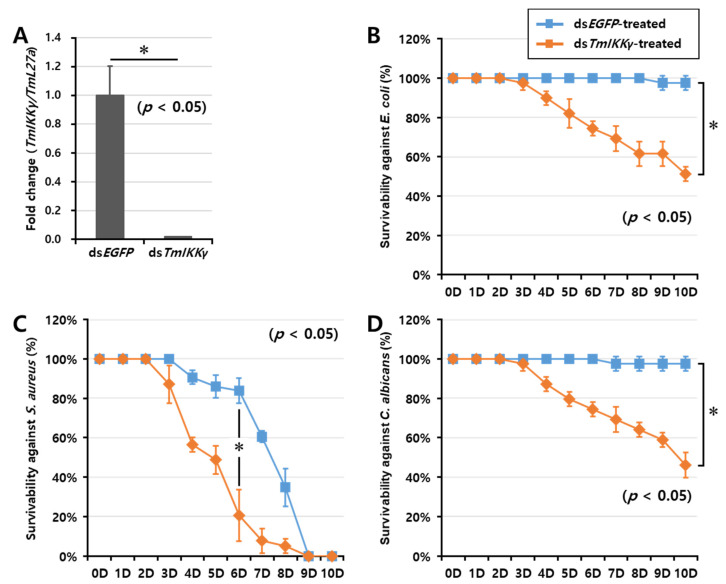
RNA interference (RNAi)-based silencing assay for *T. molitor* larval survival analysis after *E. coli*, *S. aureus*, and *C. albicans* injections. (**A**) RNAi silencing efficiency of *TmIKKγ* was about 99%. Time-dependent survival of double-stranded (ds)*TmIKKγ*-injected *T. molitor* larvae after challenge with *E. coli* (**B**), *S. aureus* (**C**), and *C. albicans* (**D**). The survival was studied for 10 d after the microbial challenge, with ds*EGFP)*-treated larvae acting as negative controls. Results are shown as an average of three independent biological replicates with standard errors. Asterisks denote significant differences at 95% confidence level (Wilcoxon-Mann Whitney test).

**Figure 5 ijms-21-06734-f005:**
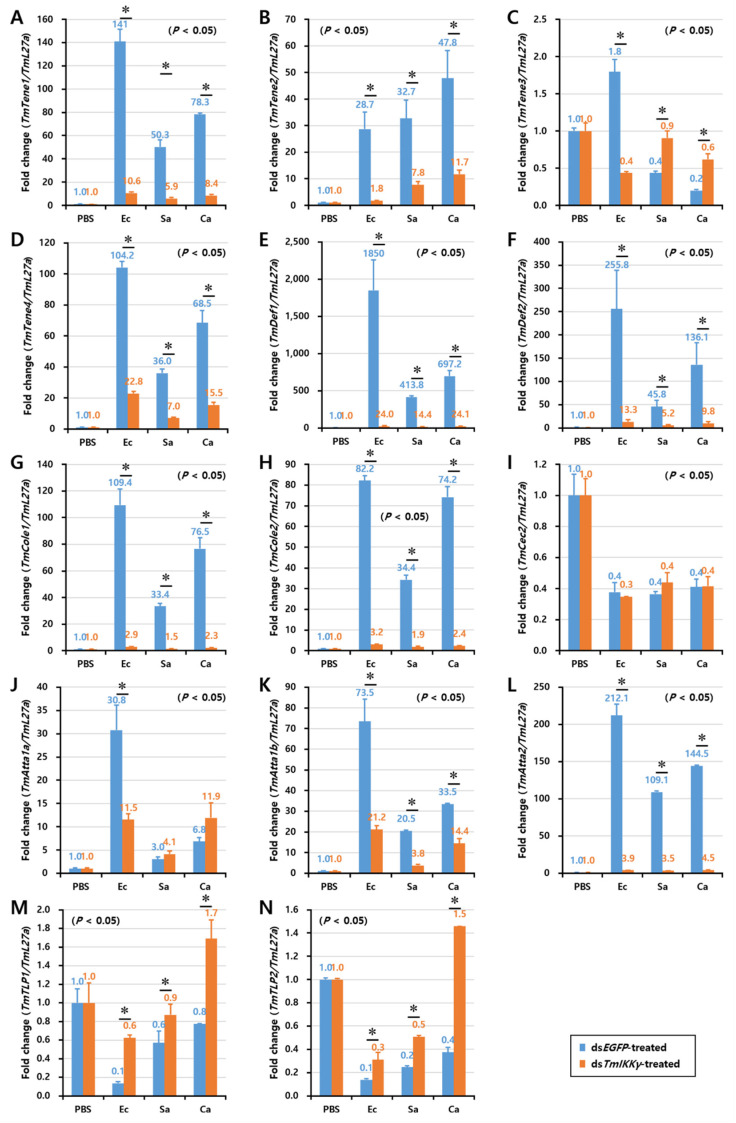
Induction patterns of fourteen antimicrobial peptide (AMP) genes in *TmIKKγ*-silenced *T. molitor* larval hemocytes in response to *E. coli*, *S. aureus*, and *C. albicans* infections. This includes *TmTene1* (**A**), *TmTene2* (**B**), *TmTene3* (**C**), *TmTene4* (**D**), *TmDef1* (**E**), *TmDef2* (**F**), *TmCole1* (**G**), *TmCole2* (**H**), *TmCec2* (**I**), *TmAtta1a* (**J**), *TmAtta1b* (**K**), *TmAtta2* (**L**), *TmTLP1* (**M**), and *TmTLP2* (**N**). The experimental samples were divided into three groups (*E. coli*, *S. aureus*, or *C. albicans*-challenged groups) and a wounded control (PBS group). AMP expression was studied 24 h after microbial challenge. ds*EGFP* was used as a negative control, and *TmL27a* served as an internal control. The numbers above the bars represent the AMP transcription levels. All experiments were repeated three times, and similar results were obtained. Asterisks denote significance at 95% confidence levels.

**Figure 6 ijms-21-06734-f006:**
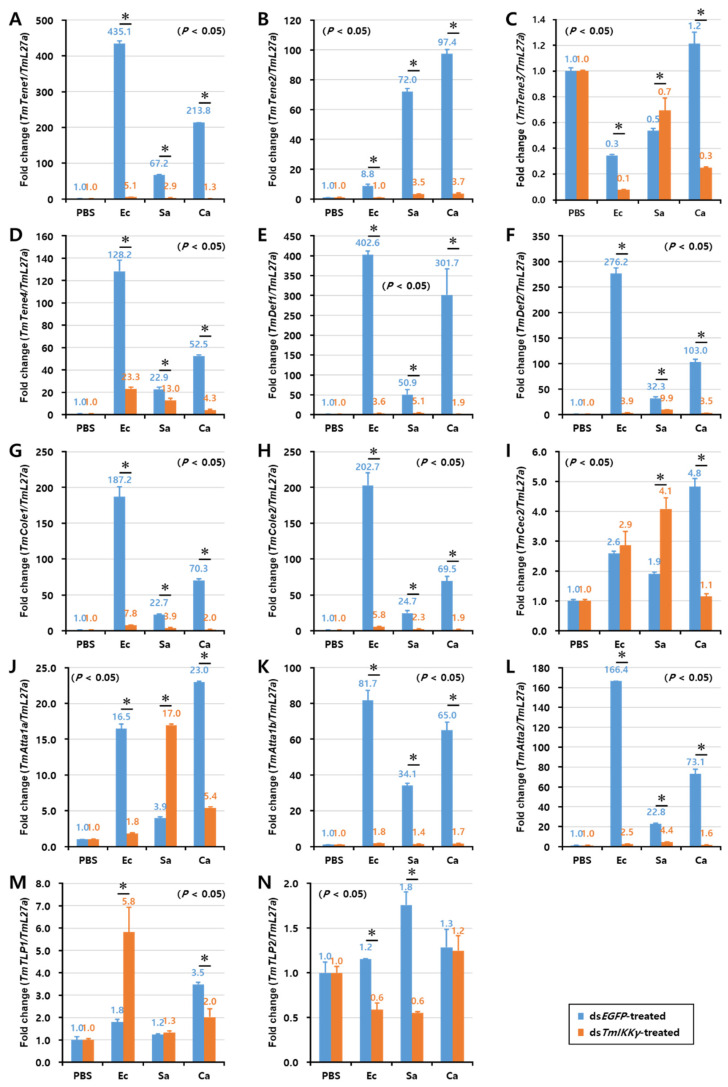
Induction patterns of fourteen antimicrobial peptide (AMP) genes in the larval gut of *TmIKKγ*-silenced *T. molitor*, in response to *E. coli*, *S. aureus*, and *C. albicans* infections. This includes *TmTene1* (**A**), *TmTene2* (**B**), *TmTene3* (**C**), *TmTene4* (**D**), *TmDef1* (**E**), *TmDef2* (**F**), *TmCole1* (**G**), *TmCole2* (**H**), *TmCec2* (**I**), *TmAtta1a* (**J**), *TmAtta1b* (**K**), *TmAtta2* (**L**), *TmTLP1* (**M**), and *TmTLP2* (**N**). The experimental samples were divided into three groups (*E. coli*, *S. aureus*, or *C. albicans*-challenged groups) and a wounded control (PBS group). AMP expression was studied 24 h after microbial challenge. ds*EGFP* was used as a negative control, and *TmL27a* served as an internal control. The numbers above the bars represent the AMP transcription levels. All experiments were repeated three times, and similar results were obtained. Asterisks denote significance at 95% confidence levels.

**Figure 7 ijms-21-06734-f007:**
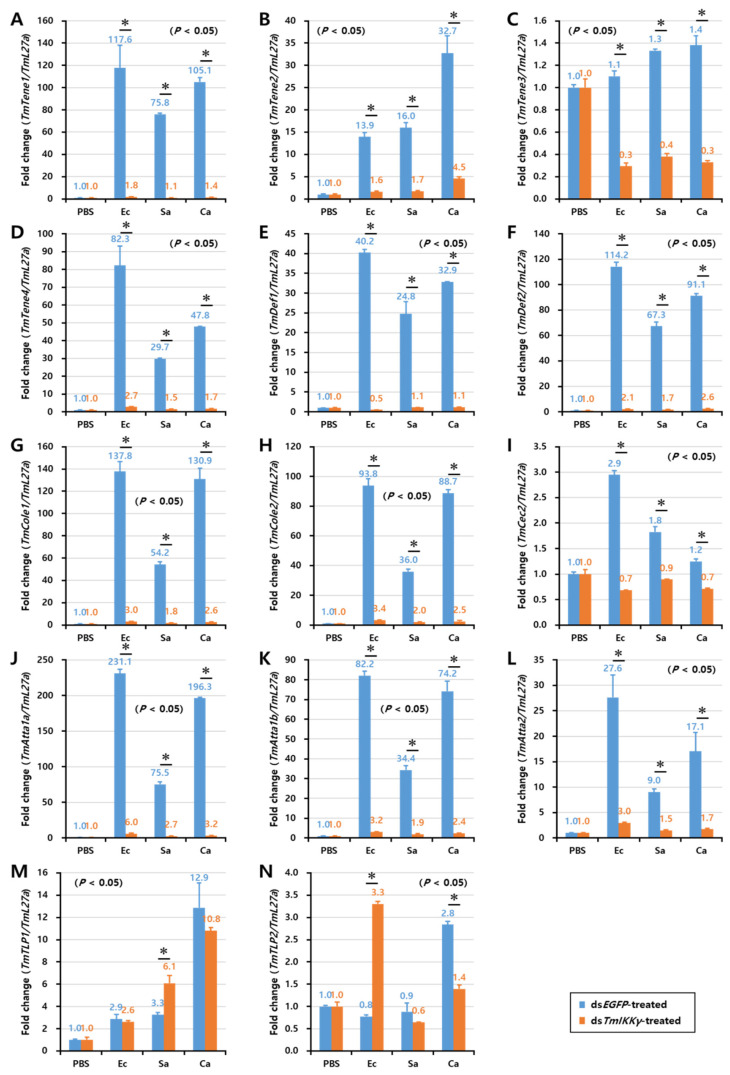
Induction patterns of fourteen antimicrobial peptide (AMP) genes in the larval fat body of *TmIKKγ*-silenced *T. molitor* in response to *E. coli*, *S. aureus*, and *C. albicans* infections. This includes *TmTene1* (**A**), *TmTene2* (**B**), *TmTene3* (**C**), *TmTene4* (**D**), *TmDef1* (**E**), *TmDef2* (**F**), *TmCole1* (**G**), *TmCole2* (**H**), *TmCec2* (**I**), *TmAtta1a* (**J**), *TmAtta1b* (**K**), *TmAtta2* (**L**), *TmTLP1* (**M**), and *TmTLP2* (**N**). The experimental samples were divided into three groups (*E. coli*, *S. aureus*, or *C. albicans*-challenged groups) and a wounded control (PBS group). AMP expression was studied 24 h after the microbial challenge. ds*EGFP* was used as a negative control, and *TmL27a* served as an internal control. The numbers above the bars represent the AMP transcription levels. All experiments were repeated three times, and similar results were obtained. Asterisks denote significance at 95% confidence levels.

**Figure 8 ijms-21-06734-f008:**
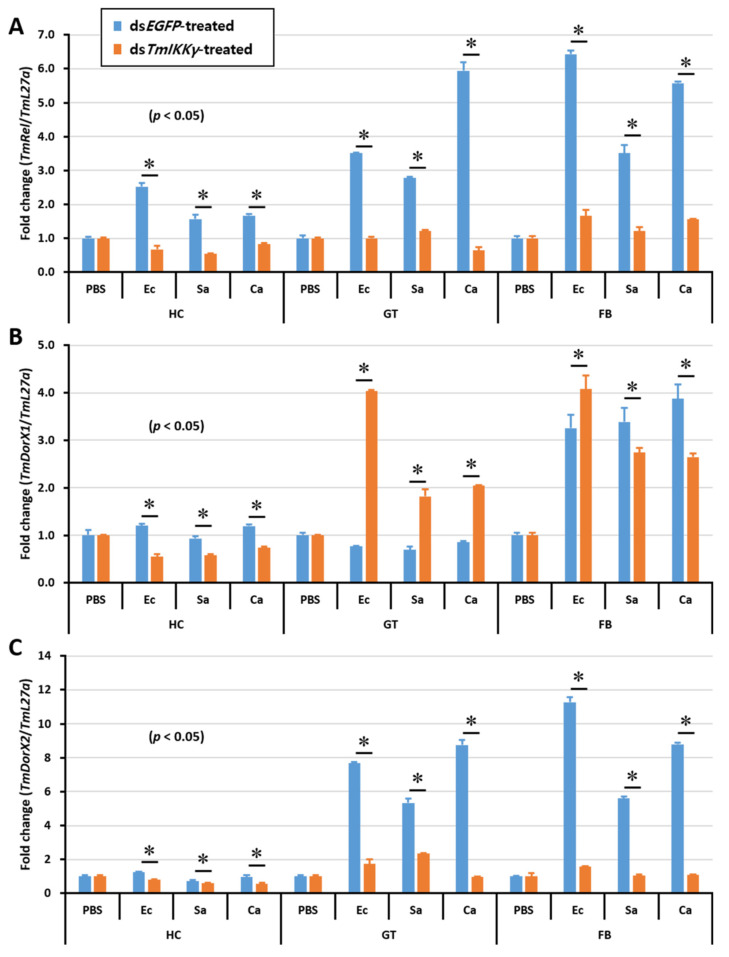
Tissue-specific induction patterns of NF-κB genes in ds*TmIKKγ*-treated *T. molitor* larvae after injections of *E. coli*, *S. aureus*, and *C. albicans*. mRNA expression levels of *TmRelish1* (**A**), *TmDorX1* (**B**), and *TmDorX2* (**C**). *T. molitor* larvae injected with PBS acted as the mock control group. ds*EGFP* was used as a negative control, and *TmL27a* served as an internal control. Asterisk denotes significant differences from the mock control at 95% confidence levels (Student’s *t*-test).

**Figure 9 ijms-21-06734-f009:**
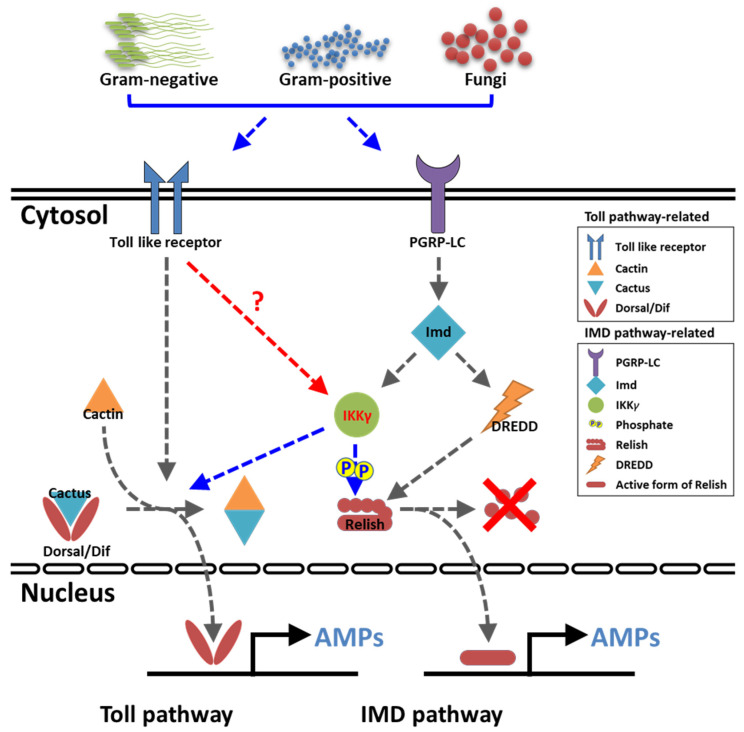
Graphical summary of *Tm*IKKγ on the AMP production in *T. molitor*. Our results suggested that *Tm*IKKγ regulates the AMP gene expression in response to three pathogens, including *E. coli*, *S. aureus*, and *C. albicans*, through both NF-κB transcription factors, *Tm*Relish and *Tm*DorX2.

**Table 1 ijms-21-06734-t001:** Primers used in this study.

Name	Primer Sequences
*TmIKKγ*_cloning_Fw	5′-TTGTTGTTCTGGGACCTTCC-3′
*TmIKKγ*_cloning_Rv	5′-GAGGGTGGTTGTTTGCAGTT-3′
*TmIKKγ*_qPCR_Fw	5′-TTGCGCAATTCCAAGGTGAC-3′
*TmIKKγ*_qPCR_Rv	5′-AGCCCCACAATTTCGGTTTG-3′
ds*TmIKKγ*_Fw	5′-TAATACGACTCACTATAGGGT
	ATTTCCAGAGGCACCAACAG-3′
ds*TmIKKγ*_Rv	5′-TAATACGACTCACTATAGGGT
	AACTTGCTCTCTGCCAGCTC-3′
ds*EGFP*_Fw	5′-TAATACGACTCACTATAGGG
	TACGTAAACGGCCACAAGTTC-3′
ds*EGFP*_Rv	5′-TAATACGACTCACTATAGGG
	TTGCTCAGGTAGTGTTGTCG-3′
*TmTenecin1*_Fw	5′-CAGCTGAAGAAATCGAACAAGG-3′
*TmTenecin1*_Rv	5′-CAGACCCTCTTTCCGTTACAGT-3′
*TmTenecin2*_Fw	5′-CAGCAAAACGGAGGATGGTC-3′
*TmTenecin2*_Rv	5′-CGTTGAAATCGTGATCTTGTCC-3′
*TmTenecin3*_Fw	5′-GATTTGCTTGATTCTGGTGGTC-3′
*TmTenecin3*_Rv	5′-CTGATGGCCTCCTAAATGTCC-3′
*TmTenecin4*_Fw	5′-GGACATTGAAGATCCAGGAAAG-3′
*TmTenecin4*_Rv	5′-CGGTGTTCCTTATGTAGAGCTG-3′
*TmDefensin1*_Fw	5′-AAATCGAACAAGGCCAACAC-3′
*TmDefencin1*_Rv	5′-GCAAATGCAGACCCTCTTTC-3′
*TmDefensin2*_Fw	5′-GGGATGCCTCATGAAGATGTAG-3′
*TmDefensin2*_Rv	5′-CCAATGCAAACACATTCGTC-3′
*TmColeoptericin1*_Fw	5′-GGACAGAATGGTGGATGGTC-3′
*TmColeoptericin1*_Rv	5′-CTCCAACATTCCAGGTAGGC-3′
*TmColeoptericin2*_Fw	5′-GGACGGTTCTGATCTTCTTGAT-3′
*TmColeoptericin2*_Rv	5′-CAGCTGTTTGTTTGTTCTCGTC-3′
*TmAttacin1a*_Fw	5′-GAAACGAAATGGAAGGTGGA-3′
*TmAttacin1a*_Rv	5′-TGCTTCGGCAGACAATACAG-3′
*TmAttacin1b*_Fw	5′-GAGCTGTGAATGCAGGACAA-3′
*TmAttacin1b*_Rv	5′-CCCTCTGATGAAACCTCCAA-3′
*TmAttacin2*_Fw	5′-AACTGGGATATTCGCACGTC-3′
*TmAttacin2*_Rv	5′-CCCTCCGAAATGTCTGTTGT-3′
*TmCecropin2*_Fw	5′-TACTAGCAGCGCCAAAACCT-3′
*TmCecropin2*_Rv	5′-CTGGAACATTAGGCGGAGAA-3′
*TmThaumatin-like protein1*_Fw	5′-CTCAAAGGACACGCAGGACT-3′
*TmThaumatin-like protein1*_Rv	5′-ACTTTGAGCTTCTCGGGACA-3′
*TmThaumatin-like protein2*_Fw	5′-CCGTCTGGCTAGGAGTTCTG-3′
*TmThaumatin-like protein2*_Rv	5′-ACTCCTCCAGCTCCGTTACA-3′
*TmDorX1*_qPCR_Fw	5′-AGCGTTGAGGTTTCGGTATG-3′
*TmDorX1*_qPCR_Rv	5′-TCTTTGGTGACGCAAGACAC-3′
*TmDorX2*_qPCR_Fw	5′-ACACCCCCGAAATCACAAAC-3′
*TmDorX2*_qPCR_Rv	5′-TTTCAGAGCGCCAGGTTTTG-3′
*TmRelish*_qPCR_Fw	5′-AGCGTCAAGTTGGAGCAGAT-3′
*TmRelish*_qPCR_Rv	5′-GTCCGGACCTCAAGTGT-3′
*TmL27a*_qPCR_Fw	5′-TCATCCTGAAGGCAAAGCTCCAGT-3′
*TmL27a*_qPCR_Rv	5′-AGGTTGGTTAGGCAGGCACCTTTA-3′

Underlined regions indicate T7 promoter sequences.

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
