# Peer review of "IKKγ/NEMO Is Required to Confer Antimicrobial Innate Immune Responses in the Yellow Mealworm, Tenebrio Molitor"

_ijms, 2020, doi:10.3390/ijms21186734_

Round 1

Reviewer 1 Report

Review of Ko et al., 2020

The authors use transcript data from Tenebrio molitor to identify sequence data corresponding to TmIKKγ, a gene that regulate IKKa and IKKb. This is a nice study of the involvement of this gene in the immunity against bacterial challenge for T. molitor.  I have a few questions that require clarification.

In the abstract, change “The TmIKKγ transcript is 1,521 bp that putatively encodes a polypeptide of 506 amino acid residues.”  Of course, you cannot say anything about the exon because you do now have the genome data.  Also, at the end of the abstract, the expression of a few more antimicrobial genes is thrown in, with no explanation of why.  This sentence in the introduction “In this study, we have functionally characterized T. molitor IKKγ (TmIKKγ), using an RNAi screen, by elucidating its role in regulating T. molitor AMP genes after microbial challenge.” Could be paraphrased in the abstract to clarify.

Line 99, the correct common name is the yellow mealworm.

Line 117-8, as far as I know, there is no T. molitor genome?  But, there are EST databases, which one did you use?  In next lines, you say “Homology mapping of TmIKKγ against T. molitor genome database revealed a single exon of 1,521 bp”.  I find it hard to believe that this gene has only one exon.  I see that the genomic data is provided in the supplementary figures – where did this come from?  How unusual is it to find only one coding exon?  Are these single exon genes in other insects?

Line 173, change to “fold”

Lines 194-96, “With S. aureus infection, the mortality trend was unclear initially, but, at day 6 post-infection, there was a significant reduction in mortality in dsTmIKKγ-treated larvae as compared with dsEGFP-treated larvae (Figure 4C).” 

Do you mean to say “. . .there was a significant increase in mortality in dsRMIKKY-treated larvae. . .” ? Also, why do you see mortality with your dsEGFP control?  Would you also see it with noninjected larvae?

In Figure 3 legend, do you mean “of T. molitor larvae following . . . “

I wanted to point out that in experimental protocol, you may save yourselves a lot of time in using RNA-Seq to look at the effect of a knockdown on gene expression, instead of qPCR.  In this way, you may also see other genes that are unexpectedly affected by the KD.

The Discussion section is complex and may be easier to understand if you divide it into more paragraphs.  For example, in line 277, start new paragraph with “In this study . . . “; maybe separate the effects of phosphorylation and ubiquination into separate sections?  Definitely need to chop up last paragraph.  May I also suggest a diagram of the very complex signaling pathway?  This could help those that are less familiar with this system understand the results and discussion easier.  After I wrote this, I see that you in fact DO have a diagram – this would have been helpful to refer to while reading the document.  Please, reference this diagram in the Introduction.

Author Response

The authors use transcript data from Tenebrio molitor to identify sequence data corresponding to TmIKKγ, a gene that regulate IKKα and IKKβ. This is a nice study of the involvement of this gene in the immunity against bacterial challenge for T. molitor.

Author’s response: The authors thank the reviewer for understanding the research in the manuscript and appreciating the work.

In the abstract change “The TmIKKγ transcript is 1,521 bp that putatively encodes a polypeptide of 506 amino acid residues.” Of course, you cannot say anything about the exon because you do not have the genome data. Also, at the end of the abstract, the expression of a few more antimicrobial genes is thrown in, with no explanation of why. This sentence in introduction “In this study, we have functionally characterized T. molitor IKKγ (TmIKKγ), using an RNAi screen, by elucidating its role in regulating T. molitor AMP genes after microbial challenge”. Could be paraphrased in the abstract to clarify.

Author’s response: We previously performed RNA sequencing as well as Genome sequencing with T. molitor, thus, genome data is available. However, it is unpublished. For understanding the gene structure of IKKγ, we conducted local tblastn analysis using TmIKKγ as query against the T. molitor transcriptome and genome database. Please find the details in Materials and Methods section (lines 397-400). The in silico representation was sufficient to screen a single exon representing TmIKKγ. To cite an earlier published article by Jo et al., 2019 in Scientific Reports (A Nature Group Publications), TmIMD gene sequence was retrieved by conducting local tblastn analysis using TmIMD as the query against the T. molitor genome database. However in this case, TmIMD showed 3 exons. We therefore confirm our procedural accuracy for this study. Nevertheless, we have changed the abstract as desired by the reviewer.

Line 99, the correct common name is the yellow mealworm.

Author’s response: Revised accordingly.

Line 117-8, as far as I know, there is no T. molitor genome? But, there are EST databases, which one did you use? In next few lines, you say “Homology mapping of TmIKKγ against the T. molitor genome database revealed a single exon of 1,521 bp”. I find it hard to believe that this gene has only one exon. I see that the genomic data is provided in the supplementary figures- where did this come from? How unusual is it to find only one coding exon? Are these single exon genes in other insects?

Author’s response: As stated in the rebuttal to an earlier query, we reconfirm that the T. molitor genome (DNA-Seq) database is available in the lab but it is unpublished. Hence, we have used the T. molitor DNA-Seq database to screen the gene organization of TmIKKγ. We have not taken any EST database as subject in this study. Hence, we confirm the genomic data provided in the supplementary figure.

Further, as pointed out, the presence of single-exon genes in intron-rich, multicellular eukaryotic genome is perplexing. As summarized by Sakharkar et al., 2004; Frontiers in Biosciences, single-exon genes fraction decreases with gene content and increases with gene density in eukaryotic genomes including the model insect D. melanogaster. Further, we believe that single-exon genes have evolved to become robust to mis-transcription avoiding fragile codons relative to robust codons. Hence, single-exon genes in genomes of insects cannot be ruled out.

Line 173, change to ‘fold’

Author’s response: Revised accordingly.

Lines 194-96, “With S. aureus infection, the mortality trend was unclear initially, but, at day 6 post-infection, there was a significant reduction in mortality in dsTmIKKγ-treated larvae as compared with dsEGFP-treated larvae (Figure 4C).”

Do you mean to say “…. There was a significant increase in mortality in dsTmIKKγ-treated larvae…”? Also, why do you see mortality with your dsEGFP control? Would you also see it with non-injected larvae?

Author’s response: We have not conducted mortality assay with non-injected larvae, but suspecting the health status of the larvae chosen in this group, we could have presumably seen very similar trends with non-injected larvae.

I wanted to point out that in experimental protocol, you may save yourselves a lot of time in using RNA-Seq to look at the effect of a knockdown on gene expression, instead of qPCR.  In this way, you may also see other genes that are unexpectedly affected by the KD.

Author’s response: We understand that RNA-Seq has been used to validate knockdown of target genes, and to examine the effect of knockdown on other genes. This has been successfully applied in the coleopteran model, T. castaneum and helps in genome-wide validation and analysis in the targeted pest chosen in the pest-control programmes.

The Discussion section is complex and may be easier to understand if you divide it into more paragraphs.  For example, in line 277, start new paragraph with “In this study . . . “; maybe separate the effects of phosphorylation and ubiquination into separate sections?  Definitely need to chop up last paragraph.  May I also suggest a diagram of the very complex signaling pathway?  This could help those that are less familiar with this system understand the results and discussion easier.  After I wrote this, I see that you in fact DO have a diagram – this would have been helpful to refer to while reading the document.  Please, reference this diagram in the Introduction.

Author’s response: As per the reviewer’s suggestions, we have divided the discussion section into paragraphs (separate sections). Further, figure 9 diagram has been cited in the ‘Introduction’ section.

Reviewer 2 Report

The study by Ko et al describes the intriguing importance of insect IKKγ in antimicrobial response and larval survival. The authors elegantly uses siRNA knockdown in Tenebrio molitor infection model to deduce the role of IKKγ in mediating NF-κB signalling, Toll and IMD pathway crosstalking to regulate expression of innate defense peptides. While I enjoyed reading the manuscript, I would like to bother the authors with a few minor concerns:

  • In Figure 3, the level of TmIKKγ expression appears to be low, even downregulated, at 24 h following any microbial infection in larval gut, fat body amd hemocytes. I wonder what was the rationale behind of choosing this timepoint for Figure 5, 6 and 7. And despite the low TmIKKγ, the expression of the innate defense peptides in dsEGFP control are quite high. Although the 12 h TmIKKγ expression is high, the high expression of defense peptides is unlikely because of this considering the very short turnover of mRNAs. This discrepancy is my biggest concern because it may effectively negate the whole concept of TmIKKγ-regulated defense response. Did the authors confirm if the TmIKKγ siRNA have any offtarget effect that may contribute to the decrease of innate defense expression?
  • The conclusion that TmIKKγ positively and negatively regulates innate response in larval gut is relatively rush, because ;ike hemocytes and fat body, TmIKKγ silencing affects the similar set of innate defense genes. Therefore, current data do not support this conclusion.
  • Also, how TmIKKγ transcriptionally regulates the NF-kB genes is not explored, discussed or speculated. Therefore, the proposed model may not be as strong as it should.
  • Have the authors considered to complement the mRNA expression data with protein expression? That would strengthen the findings.
  • Minor point: is that a mislabelling in Figure 8? The chart titles should be the gene names not “in hemocytes/gut/fat body”?

I hope the authors will find my comments helpful in improvig their manuscript. 
I eagerly look forward to reading the revised version.

Author Response

The study by Ko et al describes the intriguing importance of insect IKKγ in antimicrobial response and larval survival. The authors elegantly uses siRNA knockdown in Tenebrio molitor infection model to deduce the role of IKKγ in mediating NF-κB signaling, Toll and IMD pathway cross-talking to regulate expression of innate defense peptides. While I enjoyed reading the manuscript, I would like to bother the authors with a few minor concerns:

Author’s response: We deeply acknowledge the constructive comments from the reviewer and are happy to answer his minor concerns.

In Figure 3, the level of TmIKKγ expression appears to be low, even downregulated, at 24 h following any microbial infection in larval gut, fat body and hemocytes. I wonder what was the rationale behind of choosing this time-point for Figure 5, 6 and 7. And despite the low TmIKKγ, the expressions of the innate defense peptides in dsEGFP control are quite high. Although the 12 h TmIKKγ expression is high, the high expression of defense peptides is unlikely because of this considering the very short turnover of mRNAs. This discrepancy is my biggest concern because it may effectively negate the whole concept of TmIKKγ-regulated defense response. Did the authors confirm if the TmIKKγ siRNA have any off-target effect that may contribute to the decrease of innate defense expression?

Author’s response: It is well known that the IKKs plays critical role in regulation of NF-kB, Relish (IMD pathway) and the activation of relish results AMP production. Thus, activation of TmIKKγ gradually initiates AMP expression. In addition, for the investigation of effects of TmIKKγ on antimicrobial peptide expression, we initially confirmed the highest antimicrobial activity against various pathogens at different time points. And we finally choose the highest antimicrobial activity against pathogens at 24 h-post injection. Thus, we investigated the AMP expression at 24 h-post challenge of pathogens.

As stated in Materials and Methods section under ‘TmIKKγ RNAi’, the primers for double-stranded RNA (dsRNA) were designed using the Snapdragon software (http://www.flyrnai.org/cgi-bin/RNAi_find_primers.pl) to prevent cross-silencing effects. In addition, we initially identified three IKK genes, named IKK-γ, -β and -ε, and performed similar experiments (we will publish these dataset as soon as possible). During these experiments, we confirmed that there are no off-target effect.

The conclusion that TmIKKγ positively and negatively regulates innate response in larval gut is relatively rush, because like hemocytes and fat body, TmIKKγ silencing affects the similar set of innate defense genes. Therefore, current data do not support this conclusion.

Also, how TmIKKγ transcriptionally regulates the NF-kB genes is not explored, discussed or speculated. Therefore, the proposed model may not be as strong as it should.

Author’s response: In the conclusion, we have stated that the knockdown of TmIKKγ transcripts leads to downregulation of the NF-kB pathway transcription factors such as ‘Relish’ (IMD pathway) and ‘Dorsal’ isoform 2 (Toll pathway) after challenge with microorganisms. Hence, the model described in Figure 9 places IKKγ at the centre of both the Toll and IMD pathways regulating the transcriptional activation of AMPs. Further, from the data generated from our lab regarding the extracellular signaling pathway of T. molitor in last few years, we have been able to elucidate the signaling links of the NF-kB genes, and cross-signaling, if any. We are in a quest to connect the dots in the T. molitor NF-kB signaling cascade and future studies would unravel a stronger model for innate immune signaling in T. molitor.

Have the authors considered complementing the mRNA expression data with protein expression? That would strengthen the findings.

Author’s response: We haven’t expressed the protein to understand the induction at the protein level using Western blot. We consider our gene expression and RNA silencing methods robust enough to confirm the findings. Having said that, we thank the reviewer for suggested guidance.

Minor point: is that a mislabeling in Figure 8? The chart titles should be the gene names not “in hemocytes/gut/fat body”?

Author’s response: Thank you for your finding. We changed the mislabeling in Figure 8.

Reviewer 3 Report

This is well written manuscript on characterized T. molitor IKKγ (TmIKKγ), using an 111 RNAi screen, by elucidating its role in regulating T. molitor AMP genes after microbial challenge. The background and objectives are clearly formulated. The study seems to have been properly conducted. The results and the discussion are well presented. The article is appropriate for the publication.

Author Response

Author’s response: We deeply acknowledge the positive comments.

Round 2

Reviewer 2 Report

The authors have satisfactorily addressed my concerns. Thank you and well done on your manuscript.